# Analysis of the deterioration process of DC XLPE cable with protrusion defect based on the development of partial discharge

Yunjie Zhou[1], Shuting Yang[1], Jiamin Xu[1], Haosheng Lv[1], Jialiang Yuan[1], Baiyu Li [2]*

**1** State Grid Shanghai Cable Company, Shanghai, China, **2** Shanghai Jiao Tong University, Shanghai, China

* z1mubaiyu@sjtu.edu.cn

## Abstract

High-voltage direct current (HVDC) cables are essential for long-distance power transmission, particularly in renewable energy applications. Cross-linked polyethylene (XLPE) insulation is commonly used in these cables, but protrusion defects that occur during manufacturing can distort the electric field and initiate partial discharge (PD), accelerating insulation degradation. In this study, partial discharge experiments were conducted at 50 °C and 80 kV to investigate the behavior of internal semi-conductive protrusion defects in insulation, following methodologies aligned with relevant industry standards IEC 60270 for partial discharge measurements. This voltage condition is obtained from the previous pre-test using the same model, and can ensure that the cable can generate partial discharge under the conditions of 50°C and 80kV, but there will be no rapid deterioration of the cable leading to breakdown, which meets the needs of this experiment. The discharge process is divided into stages, and the relationship between discharge frequency, quantity, and cumulative discharge is explored. The results reveal a clear increase in discharge activity, especially in the fourth stage, which corresponds to the accelerated development of the discharge channel and impending insulation breakdown. These findings provide valuable insights into the defect's progression and highlight the risks of protrusion defects in HVDC cable insulation. This research contributes to the understanding of insulation degradation mechanisms and offers important data for improving the design, manufacturing, and maintenance of HVDC cables.

## 1. Introduction

The development of new energy technologies, such as photovoltaic power generation [1,2], has significantly increased the use of renewable energy in daily life [3–6] and its integration into the power grid [7]. This growth has sparked interest in flexible HVDC (High Voltage Direct Current) transmission, which can efficiently connect renewable

**Data availability statement:** The data that support the findings of this study are openly available in Figshare at https://figshare.com/, reference nuber :DOI:10.6084/m9.figshare.28658666

**Funding:** This project is supported by the Science and Technology Project of Shanghai Electric Power Company of State Grid, grant number 52090W230008. The funders had no role in study design, data collection and analysis, decision to publish, or preparation of the manuscript.

**Competing interests:** The authors have declared that no competing interests exist.

energy sources to the grid. As the voltage level of HVDC lines continues to rise, the safety of critical power equipment, such as transformers [8] and HVDC cables, has become a growing concern. Among these, the HVDC cable is essential for transmission, but it is prone to failure due to internal aging processes.

To mitigate cable aging, two primary approaches are adopted: enhancing insulation materials and assessing the cable's condition. Insulation performance can be improved by developing new materials [9,10] or modifying existing ones, like cross-linked polyethylene (XLPE), through techniques such as nano-doping [11–13]. On the other hand, evaluating the physical phenomena before aging faults occur can help predict cable failure. One such phenomenon is partial discharge (PD), which often results from defects in the insulation. Continuous PD accelerates aging, compromising the insulation's integrity [14,15].

Cable defects can be broadly categorized into common types, including voids, cracks, water trees, and protrusions, each of which contributes differently to the formation and progression of PD. Voids are air gaps or pockets within the insulation material, which significantly disrupt the electric field and are often the starting point for PD. Cracks result from mechanical stresses and thermal cycling, leading to insulation breakdown. Water trees are a form of defect caused by moisture ingress, which accelerates insulation degradation under high-voltage conditions. Among these, protrusions are small, localized geometric imperfections that extend from the semi-conductive layer into the insulation. These protrusions distort the electric field, creating localized stress points that can trigger partial discharges.

Among insulation defects, protrusions are small, localized geometric imperfections that extend from the semi-conductive layer into the insulation. These protrusions distort the electric field, creating localized stress points that can trigger partial discharges. As PD continues, protrusions evolve, forming arborescent discharge channels that grow over time. Once the electric field within the remaining insulation exceeds the breakdown threshold, the discharge channel can rapidly propagate, leading to insulation failure [16]. Therefore, understanding the formation and progression of these discharge channels is crucial for preventing cable breakdown.

Most studies on PD in XLPE cables under DC voltage focus on pin-plate electrode configurations, which are insufficient to model real-world cable defects [17,18]. Furthermore, the discharge channels are difficult to observe directly in situ, though PD characteristics remain strongly correlated with their development [19]. To address this issue, the behavior of protrusion defects in DC cables was investigated under controlled voltage and temperature conditions, and the variation in partial discharge characteristics was monitored. A finite element simulation model is established to simulate the electric field distribution and the development of discharge channels, based on electron avalanche theory. This model helps to explore the relationship between electric field distortion at the defect site, discharge magnitude, and channel size. The study aims to provide a new method for monitoring the stages of defect progression, offering insights into the early detection of faults in HVDC cables.

The results of this research have significant implications for the power transmission sector. Understanding early-stage cable degradation can improve the reliability

of HVDC systems, particularly in high-stress environments such as offshore wind farms or remote solar installations. This approach also enhances predictive maintenance, reducing the risk of sudden cable failures and ensuring stable power grid operation.

Moreover, the methods developed in this research can be extended to other high-voltage equipment, such as transformers, circuit breakers, and capacitors, where PD also plays a critical role in aging and failure. Early detection of PD can thus contribute to improved safety and efficiency in the broader electrical infrastructure.

The rest of this paper is organized as follows: Section 2 presents the experimental setup and methodology used to investigate partial discharge behavior in XLPE cables. Section 3 details the results of the partial discharge experiments, analyzing the stages of discharge and their implications for cable deterioration. Section 4 discusses the findings in the context of insulation degradation mechanisms, offering a comparison to existing literature. Finally, Section 5 concludes the paper, summarizing the key insights and suggesting future directions for HVDC cable design and testing.

## 2. Experimental and simulation methods

### 2.1. Sample preparation

This paper selects the use of 8.7/15 kV single-core cross-linked polyethylene cables in experiments. These cables are chosen to study the partial discharge characteristics during their evolution. The cable's insulation is identified as LS4258DCE from Borealis and the shielding layer is LE0550DC, which is compatible with the insulation. The primary parameters are detailed in Table 1.

The cable comprises a 6-layer structure, including a copper core, an inner semi-conductive shield, XLPE insulation, an outer semi-conductive shield, a copper tape shield, and an outer sheath. The protrusion defect is formed as a cone protruding from the inner semi-conductive shielding layer towards the insulation, with a height of 1 mm and a base radius of 0.2 mm, just as in Fig 1. The material of the cone is the same as that of the inner semi-conductive shielding layer. The specific production process is as follows:

1) Cut off the outer semi-conductive layer, the insulating layer and the inner semi-conductive layer successively in the middle of the cable.

**Table 1. XLPE cable specification used to test.**

| Structure | Parameters | Material/Process |
|---|---|---|
| Nominal Cross-Section | 95mm² | / |
| Conductor Outer Diameter | 11.5mm | Copper Wire Stranding |
| Inner Shielding Thickness | 0.8mm | Borealis LE0550DC |
| Insulation Thickness | 4.5mm | Borealis LS4258DCE |
| Outer Shielding Thickness | 1.2mm | Borealis LE0550DC |
| Sheath Thickness | 2mm | Semiconductive Polyethylene |

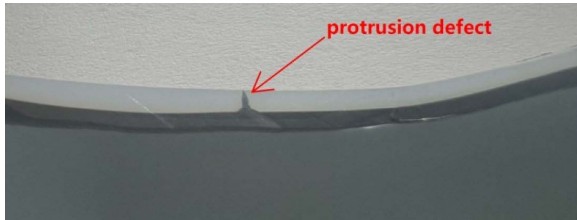

**Fig 1. Profile of protrusion defect in the cable.**

2) Then restore the inner semi-conductive layer: After vulcanization, the thickness of the restored inner semi-conductive layer is about 2 mm larger than that of the inner semi-conductive layer.

3) Cut a 1 mm protrusion in the middle of the restored inner semi-conductive layer, and scrape away the excess semi-conductive layer in the rest, that is, form the inner semi-conductive layer "protrusion".

As shown in Fig 2 and 3, the total length of the cable is 45 cm, with the defect positioned at its midpoint. To prevent surface tracking and corona discharge, 7 cm of the outer semi-conductive shielding layer was stripped from both ends of the cable to expose the XLPE insulation. Next, 3 cm of the XLPE insulation at each end was removed to expose the conductor. This configuration allowed for the insertion of a grading ring and the direct connection of the high-voltage terminal to the conductor. All exposed surfaces were carefully smoothed to minimize irregularities. At the center of the cable, a 3–5 cm wide section of the outer sheath was removed to expose the metallic shielding layer, enabling the connection of a grounding wire. To further prevent corona discharge, the entire cable and its connecting components were immersed in insulating oil.

## 2.2. Experimental circuits and methods

The high voltage circuit and partial discharge detection system for the DC cable are illustrated in Fig 4. The DC high voltage source is a model ZLT-±1200/50. The AC voltage is rectified by a high-voltage silicon stack, filtered through a 0.1-μF stabilizing capacitor to obtain DC voltage, and connected to a 100-kΩ protective resistor to prevent high currents in the event of cable breakdown. The capacitance value of the coupling capacitor is 1000 pF. To eliminate discharges from the experimental system itself, all high-voltage circuits are equipped with voltage-equalizing hoods, and the possibility of corona discharge is excluded. [20–23]

Synchronous measurements of partial discharge in cables were conducted using the impulse current method, with a measurement frequency range of 10 kHz to 1 MHz. In this study, discharges were characterized in terms of discharge quantity, measured in pico coulombs (pC), and corrected using a square wave correction module. Background noise for cables is tested under an 80-kV DC voltage condition, without artificially induced defects, did not exceed 10 pC.

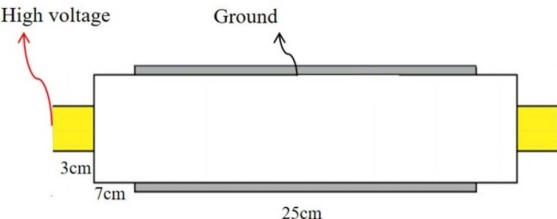

**Fig 2. Wiring diagram of the cable in schematics.**

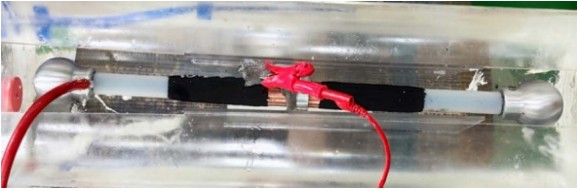

**Fig 3. Wiring diagram of the cable in Physical drawing.**

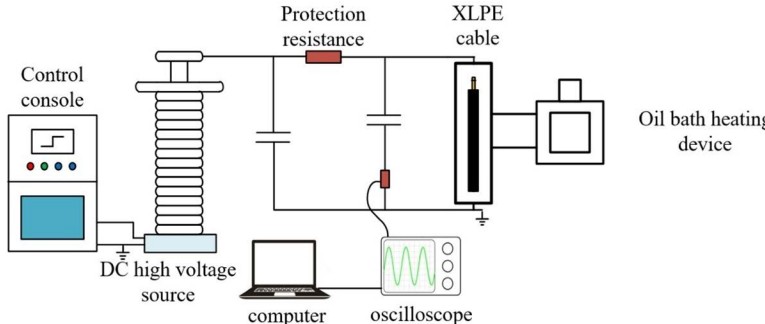

**Fig 4. The test circuit of DC cable partial discharge.**

Since the discharge characteristics of defects evolve continuously during the degradation process, this study employed an extended duration of local discharge experiments using a cyclic oil bath heating method. Given the small size of the defects and their relatively mild impact on insulation, significant defect development and discharge evolution are unlikely to occur at room temperature. However, excessively high temperatures may accelerate degradation to the point of breakdown, causing rapid and uncontrollable changes in discharge characteristics. Therefore, in this study, a cyclic oil bath heating system was used to maintain the DC cable samples at a constant temperature of 50 °C throughout the experiment.

It is important to note that this temperature does not encompass the full range of operating conditions for HVDC cables [24]. The actual operating temperature of HVDC XLPE cables depends on factors such as laying environment, ambient temperature, and current load, with long-term operating temperatures typically ranging from ambient conditions to 70 °C.

A constant voltage approach was adopted to apply voltage during testing. Selecting an appropriate voltage is crucial to accurately observe the evolution of discharge characteristics in typical defects. Excessively high voltages may cause rapid defect development and breakdown, while excessively low voltages may lead to slow defect evolution, resulting in impractically long experimental durations. Based on preliminary experiments, an applied voltage of 80 kV was determined to be optimal for studying the evolution of typical defects.

To ensure the rigor of the experiment, the experiment was carried out three times. Due to the limitation of cable technology and the randomness of partial discharge, the data of each experiment could not be the same, but the range and variation trend of discharge parameters were consistent. After collecting all the discharge waveforms of the pressure process through the oscilloscope, they are stored in the form of "txt" to the computer for processing. Because the source of abnormal data can only be electromagnetic interference, the O-coupled signal frequency and amplitude of electromagnetic interference (<10 pC) and discharge are very different. After the abnormal data are eliminated, the experimental data are sorted out, mapped, and further analysis.

## 2.3. Electric field simulation

To measure the degree of electric field distortion of protrusion defects, the defects are modeled geometrically in finite element software, and the electric field distribution is solved by an electrostatic field module. The mesh needs to be divided into smaller parts to correctly calculate because the radius of curvature at the protrusion is small. The mesh size at the protrusion is set to 2 μm. The conductor voltage is set to 80 kV. The length of the protrusion defect is 1 mm.

The only material parameters required for the physical field are relative dielectric parameters. Here, the relative dielectric constant of the semi-conductive layer is set to 20. The relative dielectric parameter of XLPE is 2.3. The relative dielectric constant of air is 1. On the boundary conditions, only the conductor voltage needs to be set to 80 kV. The outer semi-conductive layer of the cable should be connected to the ground. The steady-state study was selected.

## 2.4. Partial discharge simulation

Although the degree of electric field distortion at the defect has been understood, it is the partial discharge at the defect that really damages the insulation material. Partial discharge is a complex and constantly changing physical process. To explore what exactly happens at the microscopic level of partial discharge of the protrusion defect, the finite element simulation software was used to establish a plasma model under multiple physical fields, and the charge migration and diffusion, collision ionization, and composite adhesion behaviors were analized during the discharge process.

There is a set of hydrodynamic gas equations to describe the partial discharge process of the air gap mode [25]. This gas model has been widely used to simulate the microscopic simulation of closed gas discharge and has been proven to be effective [26].

$$\frac{\partial n_e}{\partial t} + \nabla \cdot (\mu_e \vec{E} * n_e - D_e \nabla n_e) = R_e \tag{1}$$

$$\frac{\partial n_p}{\partial t} + \nabla \cdot (\mu_p \vec{E} * n_p - D_p \nabla n_p) = R_p \tag{2}$$

$$\frac{\partial n_n}{\partial t} + \nabla \cdot (\mu_n \vec{E} * n_n - D_n \nabla n_n) = R_n \tag{3}$$

$$\nabla^2 V = \frac{-e(n_p - n_e - n_n)}{\varepsilon_r * \varepsilon_0} \tag{4}$$

Here $n_e$, $n_p$, and $n_n$ represent the charge densities of electrons, positive ions, and negative ions, respectively, with units of C/m3. $\mu_e$, $\mu_p$, and $\mu_n$ are the mobilities of electrons, positive ions, and negative ions under an electric field, respectively, with units of m²/(V·s). $\vec{E}$ is the electric field intensity vector, with units of V/m. $D_e$, $D_p$, and $D_n$ are the diffusion coefficients of electrons, positive ions, and negative ions driven by concentration gradients, respectively, with units of m²/s. $R_e$, $R_p$, and $R_n$ represent the source terms for electrons, positive ions, and negative ions, which describe the generation and annihilation of charged particles, with units of C/(m³·s), and V is the electric potential, with units of V. $\varepsilon_r$ is the relative dielectric constant; $\varepsilon_0$ is the dielectric constant of vacuum; $e$ is the elementary charge of an electron.

Equations (1~3) show the variation process of the spatial concentration of charged particles. Taking Formula (2) as an example, it is explained in detail that $\frac{\partial n_p}{\partial t}$ represents the concentration change of positive ions over time, $\nabla \cdot (\mu_p \vec{E} * n_p - D_p \nabla n_p)$ represents the change of positive ions due to migration and diffusion, and $R_p$ is the difference between the rate of positive ions generation and disappearance. In addition, considering that the migration and dissipation of charged particles can lead to changes in the electric field, this is combined with Poisson's equation (4).

$$R_e = \left| \mu_e \vec{E} \right| * n_e * \alpha - \left| \mu_e \vec{E} \right| * textn_e * \eta - \beta_{ep} n_e n_p \tag{5}$$

$$R_p = \left| \mu_e \vec{E} \right| * n_e * \alpha - \beta_{np} n_n n_p - \beta_{ep} n_e n_p \tag{6}$$

$$R_n = \left| \mu_e \vec{E} \right| * n_e * \eta - \beta_{np} n_n n_p \tag{7}$$

α is the ionization coefficient, where the impact ionization coefficient is taken; η is the adsorption coefficient; $\beta_{ep}$ and $\beta_{np}$ are the recombination coefficients of electron and positive ion and positive and negative ion respectively. In the above equations, α is the ionization coefficient (unit: 1/m), which here refers to the collision ionization coefficient. η is the attachment coefficient (unit: 1/m). $\beta_{ep}$ and $\beta_{np}$ are the recombination coefficients of electrons with positive ions and positive ions with negative ions, respectively, with units of m³/s. The ionization term $\left| \mu_e \vec{E} \right| * n_e * \alpha$ represents the process of generating new electrons and positive ions through ionization. This process is related to the electron density, ionization coefficient, electron mobility, and electric field strength. The attachment term $\left| \mu_e \vec{E} \right| * n_e * \eta$ describes the process in which electrons attach to neutral molecules to form negative ions. This term accounts for one possible mode of electron dissipation as well as the generation of negative ions. It is influenced by the electron density, attachment coefficient, electron mobility, and electric field strength. The recombination terms $\beta_{ep} n_e n_p$ and $\beta_{np} n_n n_p$ represent the decrease in the charged particle density due to recombination phenomena. The former is caused by the recombination of electrons with positive ions, while the latter results from the recombination of negative ions with positive ions. These terms are related to the densities of the recombining charged particles and are primarily determined by the recombination coefficients.

The generation and disappearance of charged particles are described by the equation of source term (5~7). In the above equation, the specific values of the above simulation parameters were obtained from the literature [27,28].

The schematic diagram of the two-dimensional geometric model is shown in Fig 5, and the geometric dimensions are consistent with those in Table 1. Free triangular mesh is selected for grid division because the time scale of gas discharge migration is small and it is greatly affected by an electric field, it needs to be divided very finely to ensure that the charge migration process of the simulation process can be accurately completed. The defect cavity was calibrated to a hydrodynamic partition, the mesh size was "extremely fine", and the maximum cell size was set to 2 μm, containing a total of 48882 in-domain cells. The division diagram is shown in Fig 6.

In the simulation, to compare the microscopic discharge process in discharge channels of different lengths, the length of discharge channels is 0.5 mm and 1 mm, and the rated conductor voltage is 15 kV. Because the time scale of charged particles migration in gas discharge process is very small, the charge injection and withdrawal at gas-solid interface can be ignored in this process. In the setting of initial conditions, this paper uses a two-dimensional Gaussian distribution array to model the initial particles (electrons and positive ions) [29]. It is expressed as follows:

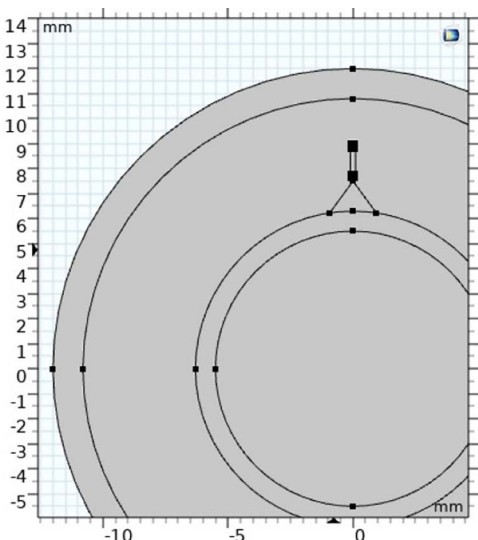

**Fig 5. Schematic diagram of discharge simulation.**

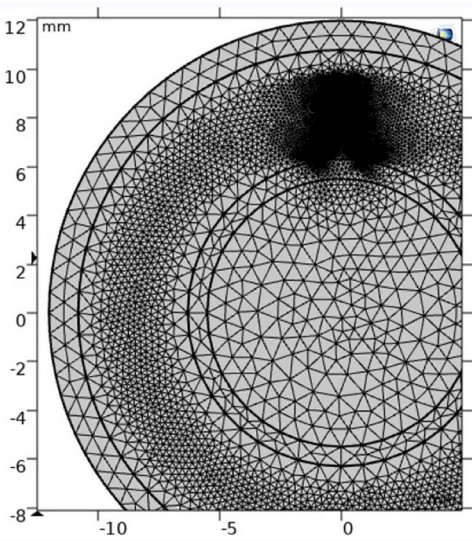

**Fig 6. Schematic diagram of discharge grid division.**

$$n_m = N_m * e^{\frac{(X-X0)^2 + (Y-Y0)^2}{-2\sigma^2}} \tag{8}$$

where $n_m$ represents the concentration distribution of the initial charged particles, and $N_m$ is the peak concentration of charged particles, $X_0$ and $Y_0$ are the horizontal and vertical coordinates of the generated initially charged particles, and $\sigma$ is the standard deviation and determines the rate at which the initial particles concentration decays with distance.

To provide a clearer understanding of the logical flow of this article, a flowchart is presented in Fig 7. This flowchart summarizes the overall logical structure and progression of the content: it begins with the introduction section, which outlines the research background and issues, as well as clarifies the research objectives; next, the methods section elaborates on the research design and experimental approaches; then, the results section presents the analyzed data and key findings; following that, the discussion section interprets the significance of the research results and compares them with existing literature; finally, the conclusion section summarizes the research contributions and offers prospects for future work.

## 3. Results and analysis

### 3.1. Experimental results and analysis

Take the discharge characteristics analysis of a representative experiment involving a protrusion defect in the inner shielding layer under positive polarity voltage as an example.

The entire discharge process is illustrated in the discharge spectrum shown in Fig 8. The discharge duration is divided into four stages: the first stage occurs during the discharge time from 0 to 40 minutes; the second stage takes place during the voltage application time from 41 to 110 minutes; the third stage occurs during the voltage application time from 111 to 140 minutes, and the fourth stage transpires during the voltage application time from 141 to 156 minutes. The basis of the discharge stage division is a sudden change in the trend of discharge frequency and discharge quantity.

As shown in the discharge spectrum of the entire experimental process with a protrusion defect in Fig 9, during the voltage application process, the distribution range of discharge quantities falls between 10 and 300 pC, with the maximum discharge quantity even reaching above 500 pC. The overall discharge quantity exhibits a clear upward trend.

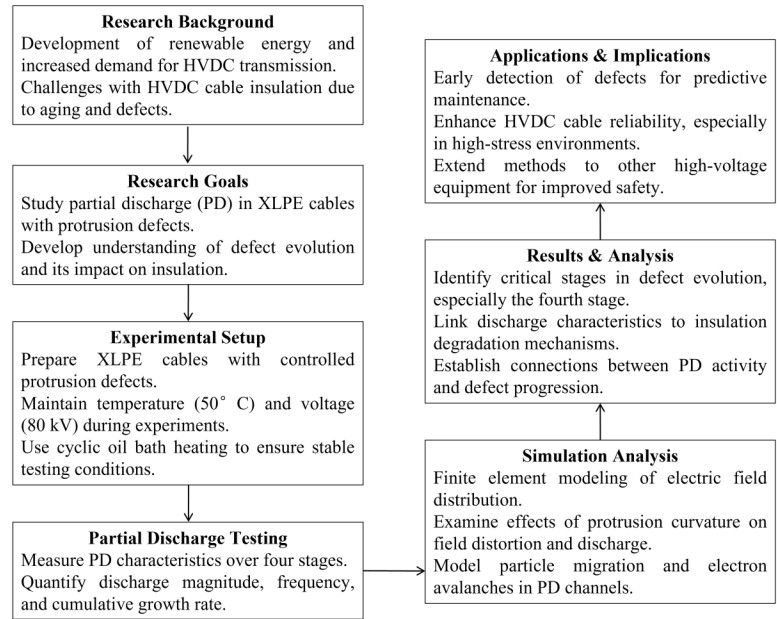

**Fig 7. Experimental workflow diagram.**

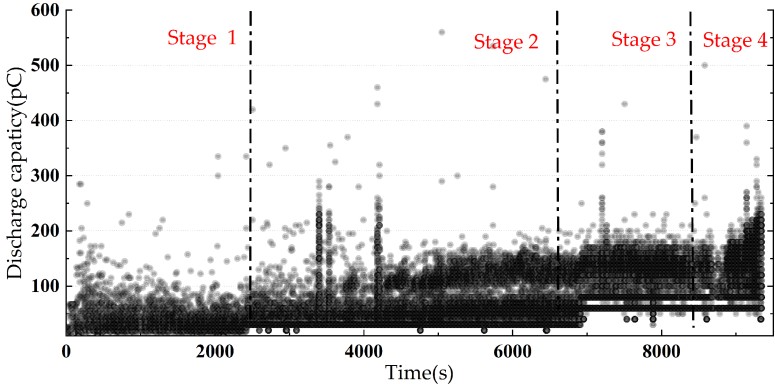

**Fig 8. The discharge pattern of the whole experiment process of the protrusion defect.**

The average discharge quantity for the entire process of the protrusion defect experiment is illustrated in Fig 9, while the discharge frequency for the entire process is depicted in Fig 10.

The average discharge quantity is the arithmetic average of the apparent discharge quantity collected per minute. The discharge frequency is the number of discharges collected per minute. In the first stage of discharge, the discharge frequency is relatively low, approximately 60 events per minute, with minimal variation. The average discharge quantity per minute shows slight fluctuations, mostly ranging between 30 and 50 pC. In the second stage of discharge, the discharge frequency starts to increase, reaching up to 200 events per minute. Simultaneously, the average discharge quantity of local discharges further rises from around 50 pC at 40 minutes to approximately 80 pC at 110 minutes. In the third stage of discharge, the discharge frequency slightly increases to about 300 events per minute, and the average discharge quantity continues to rise to 100 pC. In the fourth stage, following a brief drop in discharge frequency, the rate accelerates

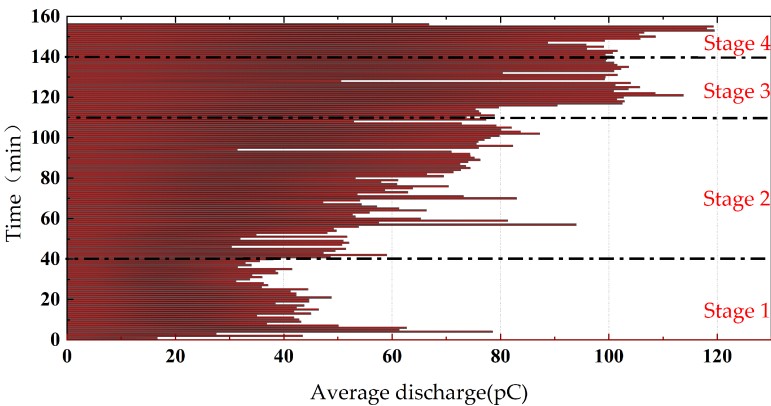

**Fig 9. The average discharge of the whole experiment process of the protrusion defect.**

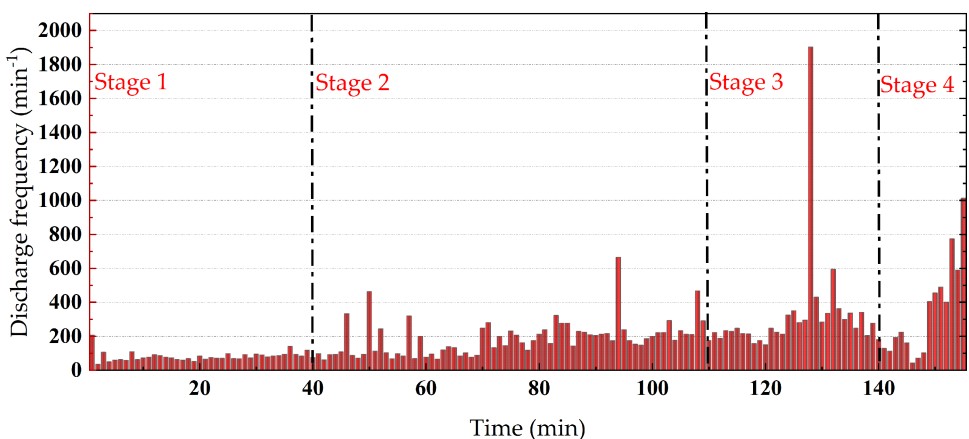

**Fig 10. Discharge frequency of the whole experiment process of the protrusion defect.**

sharply, reaching 600–1000 events per minute. Simultaneously, the average discharge quantity rises significantly to around 120 pC—more than three times its initial value. As is shown in Fig 11, during this process, the average discharge quantity further rises to around 120 pC, more than three times the initial discharge quantity.

It is worth noting that at 127 minutes, which is also the late third stage of discharge, the discharge frequency reaches a maximum of 1902 times per minute during the voltage application process. This is because, with the increase of voltage withstand time, the discharge frequency shows an upward trend in general. However, in the fourth stage just before the defect breakdown, the rapid development of the discharge channel leads to a decrease of the charge space density, which makes it difficult to achieve the condition of electron collapse, and the discharge frequency will decrease. Therefore, the maximum discharge frequency will appear in the middle and late 127 minutes of the third discharge stage. Surprisingly, the average discharge quantity decreases from the previous level of around 100 pC to 50 pC. This might be attributed to the presence of small-amplitude, densely occurring discharges, which have a slight lowering effect on the average discharge quantity.

Further analysis of the cumulative discharge quantity and its growth rate over time during the discharge process of the protrusion defect is depicted in Fig 12.

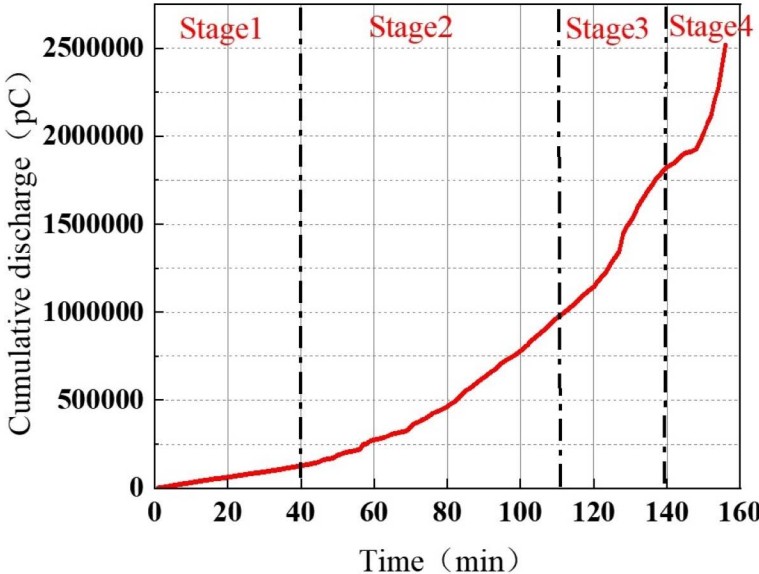

**Fig 11. The cumulative discharge quantity of the whole experiment process of the protrusion defect.**

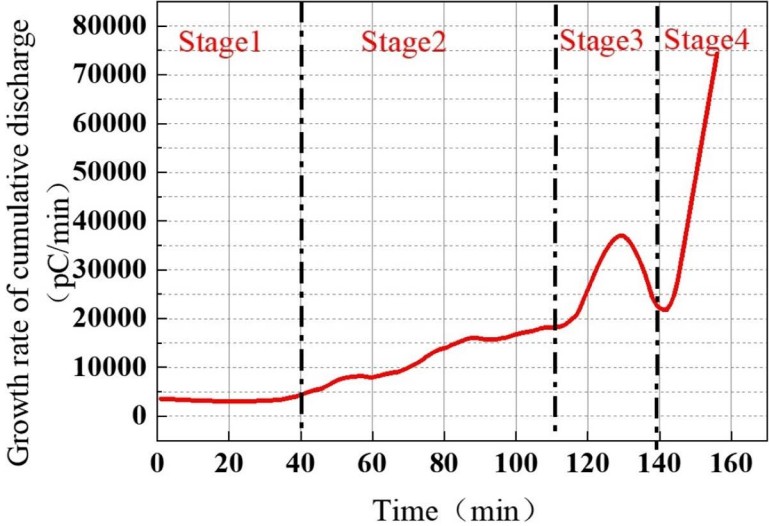

**Fig 12. The growth rate of the whole experiment process of the protrusion defect.**

In the first stage of discharge, the cumulative discharge quantity exhibits slow growth, with a relatively low growth rate, indicating limited energy injection into the defect. As the discharge progresses into the second stage, both the cumulative discharge quantity and its growth rate gradually increase, reaching approximately 1000 nC at 110 minutes. At this stage, discharge-induced damage begins to accumulate in the solid dielectric surrounding the protrusion defect. During the third stage, the cumulative discharge quantity continues to rise, approaching 2000 nC. Simultaneously, the growth rate peaks, coinciding with the maximum discharge frequency observed at 127 minutes. This suggests that the defect is undergoing accelerated development, leading to further insulation degradation. In the fourth stage, the cumulative discharge quantity

increases sharply, along with a significant rise in its growth rate, reaching 70000 pC/min. This dramatic surge indicates a substantial release of energy, highlighting severe insulation deterioration and a high risk of breakdown.

### 3.2. Simulation results and analysis

Fig 13 and 14 shows the overall distribution of the electric field in the cross-section direction from the inner semi-conductive layer to the raised protrusion defect of the cable insulation layer.

The curvature radius at the tip will affect the degree of electric field distortion. Here the radius of curvature is set for two cases, 50 μm and 10 μm respectively. It can be seen that the electric field strength in most areas of the cable insulation layer is below 30 kV/mm. However, the insulation near the tip has a very serious electric field distortion. It reaches 131 kV/mm at a radius of curvature of 50 μm. When the radius of curvature is reduced to 10 μm, it can reach nearly twice the radius of curvature of 50 μm, which is 256 kV/mm.

Fig 15 and 16 shows the radial electric field intensity distribution through the center of the bottom circle of the protrusion defect. The electric field intensity rapidly declines after reaching its maximum value at the tip and gradually becomes stable at the insulation layer near the outer semi-conductive layer. It is worth mentioning that the reduction of curvature radius not only affects the electric field intensity at the tip, but also increases the electric field strength of the insulating layer near the semi-conductive layer. In general, the curvature radius will seriously affect the degree of electric field distortion.

Through the simulation of the charge in the discharge channel, the changes in the migration and diffusion processes of electrons during a single local discharge are illustrated in Fig 17. The discharge channel has a width of 150 μm and a length of 500 μm, as shown in the fig. Following the initiation of the discharge, the migration and diffusion processes of electrons, coupled with the combination with positive ions, constitute the predominant part of the reaction. They progress toward the direction of high electric field intensity. At 1 ns, with the development of electron avalanche, the electron density at the head has reached nearly 60 times that at the beginning. By 2.1 ns, the head of the electron avalanche

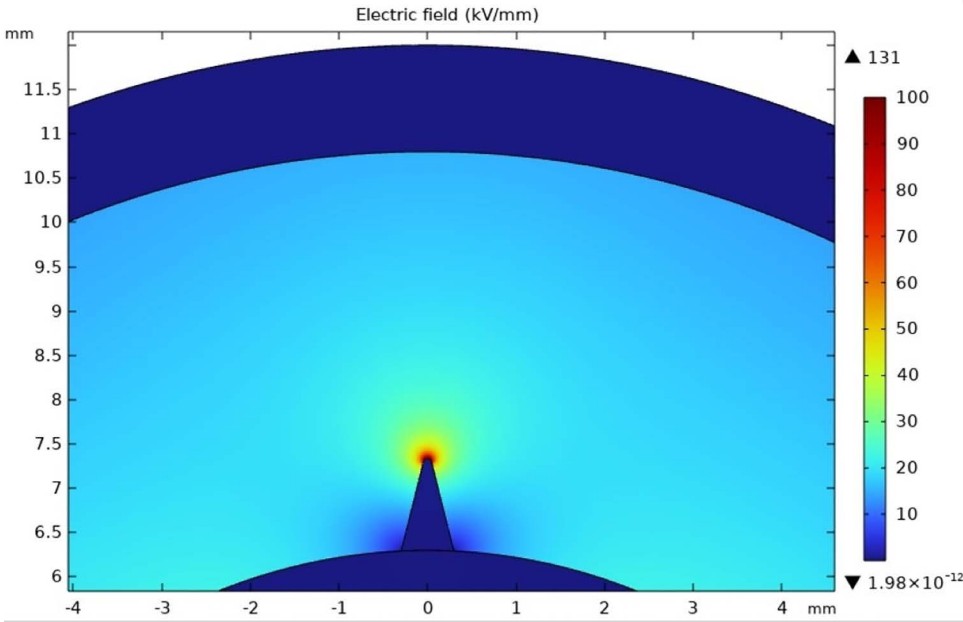

**Fig 13. The cumulative discharge quantity and its growth rate of the whole experiment process of the protrusion defect-50 μm.**

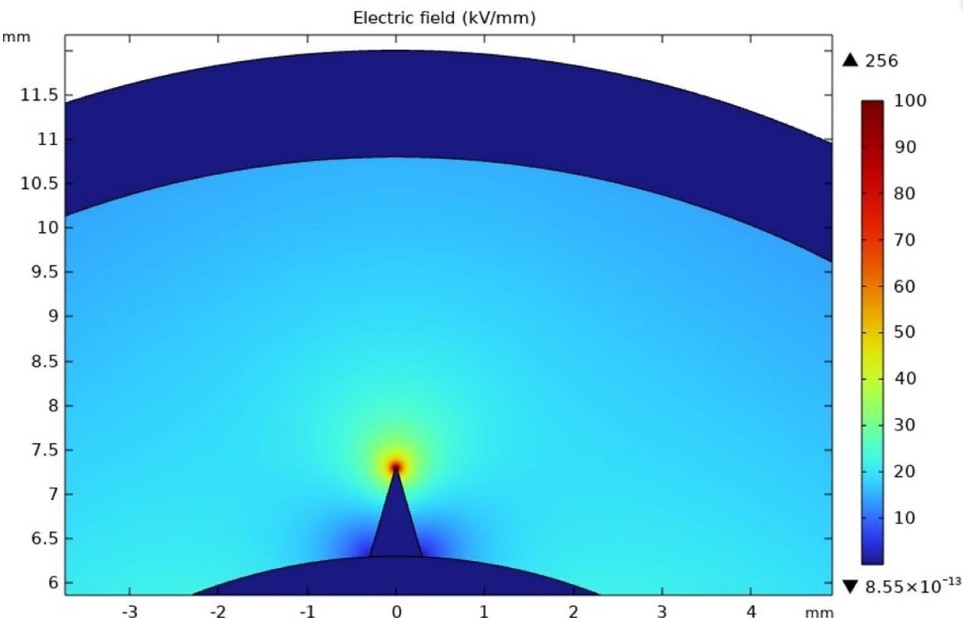

**Fig 14. The cumulative discharge quantity and its growth rate of the whole experiment process of the protrusion defect-10 μm.**

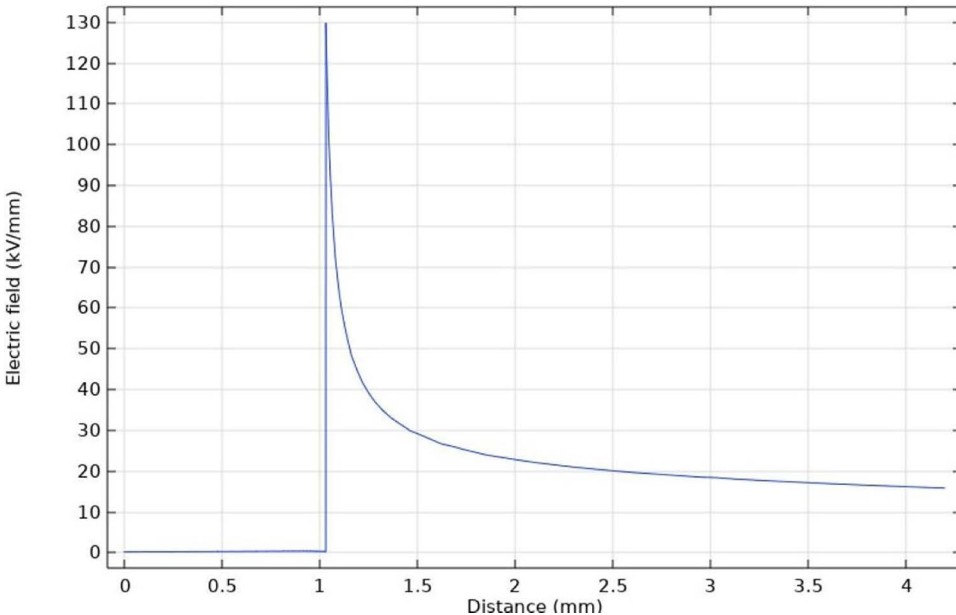

**Fig 15. Radial electric field intensity of protrusion defect with a curvature radius of 50 μm.**

has almost reached the surface of the dielectric. Electrons undergo inelastic collisions with the dielectric surface, and their energy is insufficient to continue collision ionization. At 2.5 ns, the process of electron avalanche essentially concludes.

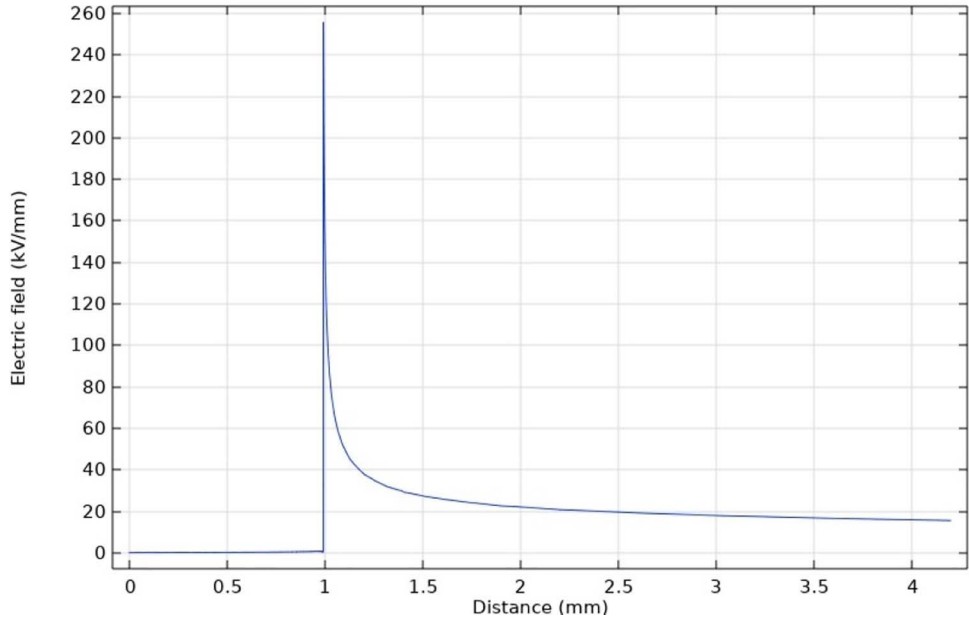

**Fig 16. Radial electric field intensity of protrusion defect with a curvature radius of 10 μm.**

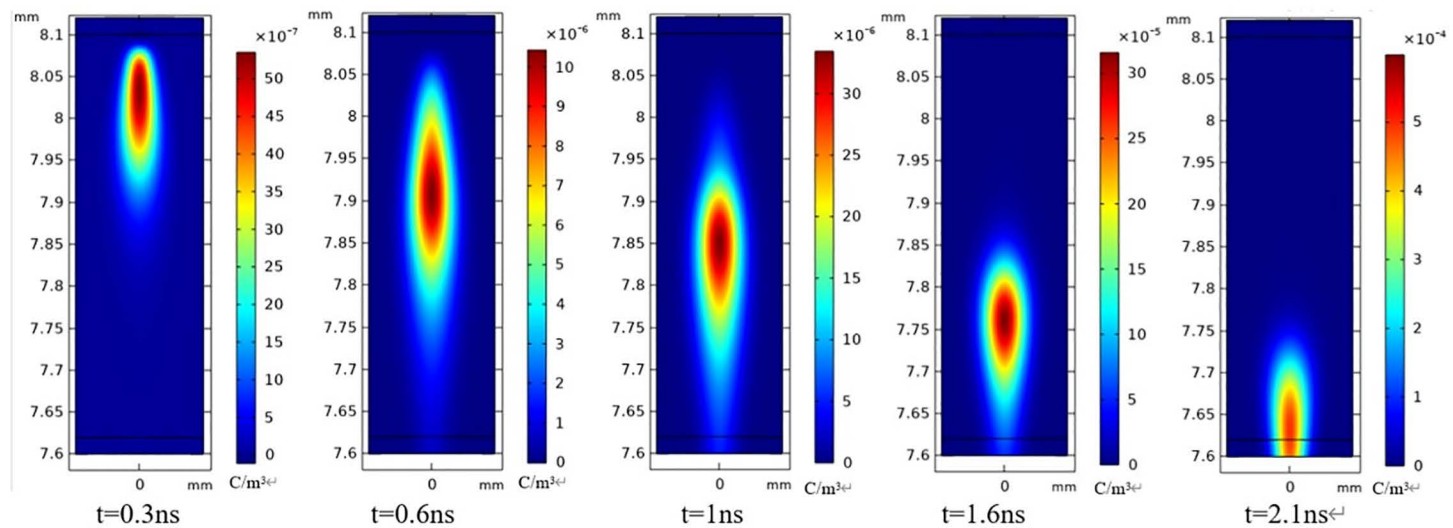

**Fig 17. Electron migration diagram of the protrusion defect's partial discharge.**

Two points need to be pointed out. Firstly, the voltage on the air gap is not decreased to 0, but to an extinction voltage [30]. In addition, there are still a lot of residual charges on the void wall, and the reverse electric field formed by these residual charges also plays a role in weakening the discharge.

## 4. Discussion

### 4.1. Physical process of protrusion defect discharge

For protrusion defects, when the applied voltage is unipolar, the charge injected into the polymer at the tip of the protrusion will be trapped by the trap, and the energy released in this process will be transferred to electrons through resonance, prompting the formation of hot electrons, and then the hot electrons will collide with the polymer molecular chain, resulting in chain breakage. At the same time, the free radical generated by the molecular chain break has a certain catalytic effect on the degradation of polymer, which leads to the expansion of the degradation range of polymer with the participation of oxygen, and promotes the formation of low density region. The increase of electron free travel in the low density region promotes the occurrence of collision ionization. The energy released by collision ionization further leads to polymer degradation in a wider range. In addition, because the electric field distortion at the tip is the most serious, the field strength is also the largest, and it will bear a large electro-mechanical stress, further accelerating the fracture of the XLPE molecular chain, and resulting in small cracks. Partial discharge is possible in such small cracks.

There are two conditions for partial discharge: the electric field strength in a certain place inside the defect reaches the minimum discharge field strength and the initial electron appears which can stimulate the electron collapse. When the partial discharge condition is reached, the discharge begins, the electrons form electron collapse, and the migration is quickly completed. In the migration process, some electrons combine with neutral particles to form negative ions. At the same time, due to the collision ionization in the process of electron migration, a large number of positive and negative ions will drift in the opposite direction under the action of the applied electric field and finally accumulate in the interface between the gas and the solid dielectric. The surface charge that accumulates at this interface generates a reverse electric field $E_r$, which reduces the overall electric field $E$ in the gas cavity and ultimately results in the termination of the discharge when the field intensity is lower than the extinction field. During the discharge interval, the surface charge will gradually decay due to the conduction along the gas-solid interface and the neutralization with gas ions, and the reverse field intensity will decrease, and the electric field in the gas cavity will gradually recover. Until the conditions for partial discharge are reached again, the next discharge begins.

It is worth noting that in this process, the migration of both electrons and positive and negative ions is completed within a few ns, which is far less than the minimum discharge interval of 0.1 ms. Therefore, the actual determining factor of the discharge interval is the speed of surface charge decay and the discharge delay. The discharge delay here refers to the moment when the minimum discharge field intensity is exceeded and the time difference when partial discharge occurs.

If the external high voltage is maintained for enough time, the air gap at the tip of the protrusion defect will continue to develop under continuous partial discharge. The final result of this process is the development of electrical tree defects, which eventually lead to the breakdown of cable insulation.

### 4.2. Evolution mechanism analysis of protrusion defect discharge

As shown in the above analysis, when a small air gap is generated near the prominent defect location due to charge injection and a large degree of electric field distortion, the air gap will gradually become larger under partial discharge, and this process is agreed to accompany the development of partial discharge. Therefore, the relationship between discharge capacity development and defects is explored respectively from the numerical calculation and the three-capacitance model.

The calculation is based on the simulation of 2.4: since the whole process of charge transfer of a discharge has been numerically simulated in 2.4. And the essence of electricity is the movement of charged particles. Therefore, the pulse current of partial discharge can be calculated by calculating the power of the electric field force of the charged particles in a discharge divided by the voltage at both ends. Just according to the following formula [31]:

$$I = \frac{e}{U_a} \iiint (n_p W_p - n_e W_e - n_n W_n)*EdV \tag{9}$$

$$W_{e,p,n} = \vec{E} * \mu_{e,p,n} \tag{10}$$

where I is the discharge current, e is the electron charge, $U_a$ is the voltage at both ends of the channel in the simulation process, $\vec{E}$ is the electric field strength, and $n_{e,p,n}$ is the concentration of charged particles. $\mu_{e,p,n}$ is the mobility of charged particles. $W_{e,p,n}$ represents the work done by the electric field force on the particles.

It is worth noting that the specific values of the physical quantities presented here can be obtained by calculating the simulation model in Part 2.4.

The length of the air domain calculated in the geometric part of the Part 2.4 model is modified from 0.5 mm to 1 mm, which means that the discharge channel is extended from 0.5 mm to 1 mm. This gives the result of Fig 18: the time to obtain the maximum discharge current also increases from 1.9 ns to 4.3 ns. At the same time, the maximum discharge current increased from 12.05 nA to 19.55 nA.

It can be seen from the simulation that the length of the discharge path greatly affects the discharge quantity, the mechanism is explored below.

The equivalent capacitance model of the discharge channel with protrusion defect in the DC field is established [32], just as in Fig 19. The capacitance at the defect of the cable is $C_3$ and the leakage resistance is $R_3$, $C_1$, $C_2$, and $R_1$, $R_2$ are the capacitance and leakage resistance of the part of the insulation in parallel and series with the defect, respectively. U is the voltage at both ends of the sample.

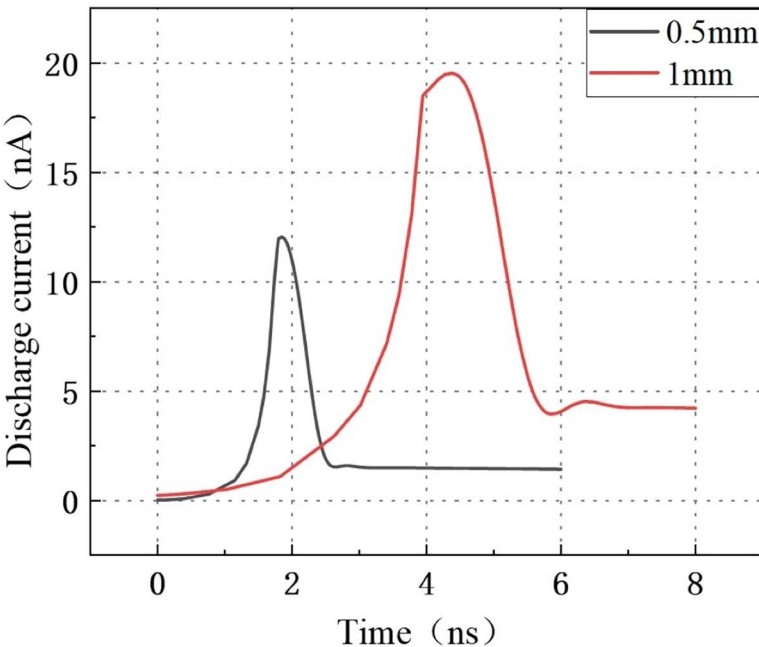

**Fig 18. Discharge currents for 0.5 mm and 1 mm discharge paths.**

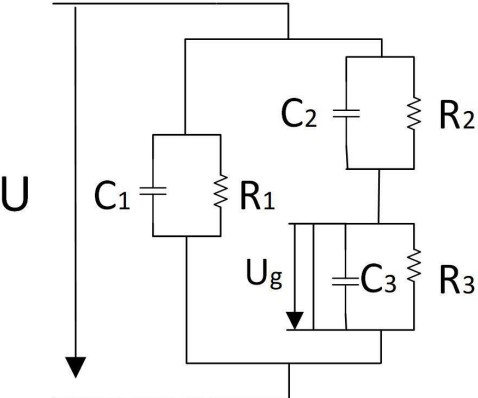

**Fig 19. Equivalent circuit for partial discharge of cable defects.**

When the defect is discharged immediately, the instantaneous voltage drop of $U_g$ is generated at the defect, and the electric field is alternating electric field at this time, the leakage resistance in the circuit can be ignored, and the voltage variation is distributed on $C_1$ and $C_2$. Apparent discharge quantity Q should meet:

$$Q = C_2 U_g$$
$$= C_2 E_b l \tag{11}$$

here, $E_b$ is the average breakdown strength in the channel during discharge; $l$ is the length of the discharge path of the defect.

Therefore, it can be concluded that the apparent discharge capacity is only determined by the channel length $l$ and the internal structure of the cable. This insulation defect will worsen in long-term continuous discharge, extending the discharge channel, and the longer discharge channel will in turn increase the discharge energy. Finally, the evolution of discharge characteristics is accelerated, and the expansion of defects is accelerated until the defects penetrate the insulation layer, making the defects completely ineffective.

## 4.3. Analysis of protrusion defect deterioration process

One step further, analysis of the development of defects based on the changes in discharge characteristics during different discharge stages. Fig 20 and 21 present the average discharge frequency and average discharge quantity of the protrusion defect at different stages. It can be observed from the averaged discharge frequency and quantity that, with the progression of discharge stages for the protrusion defect, both the averaged discharge frequency and quantity consistently increase. Specifically, the average discharge frequency rises from 82 events per minute in the first stage to 438 events per minute in the fourth stage, while the average discharge quantity increases from 41 pC in the first stage to 102 pC in the fourth stage. This changing trend aligns with the typical variation in discharge frequency and quantity observed during the DC treeing process under needle-plane electrodes. Although the protrusion of the protrusion defect in this study does not serve as a metal electrode, it may still induce electric treeing.

It is speculated that, under the influence of the applied electric field, the semi-conductive protrusion is likely to inject charges into the interior of the insulation, which is subsequently scattered and trapped after multiple collisions. This process is accompanied by the release of energy. The energy released from charge trapping is transferred to electrons in the conduction band in a non-radiative form, converting them into hot electrons [33]. The impact of hot electrons on polymer molecular chains leads to their rupture, forming electric tree channels [34,35]. Once an electric tree channel is formed, high-frequency localized discharges within this channel continually bombard the cross-linked polyethylene molecular

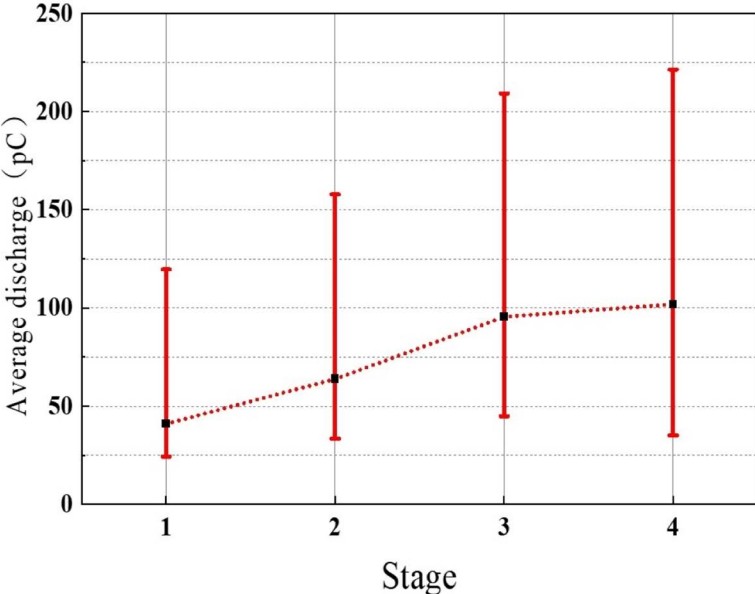

**Fig 20. Average discharge of the protrusion defect at different stages.**

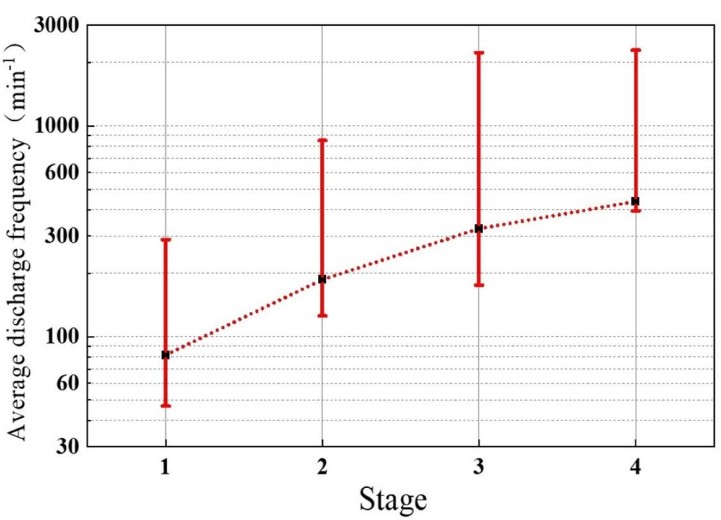

**Fig 21. Average frequency of the protrusion defect at different stages.**

chains ahead, resulting in the growth of the electric tree [36]. Longer discharge paths imply more collision ionization and electron avalanches, leading to an increase in discharge quantity and greater discharge energy. This is consistent with the sharp increase in cumulative discharge quantity and its rate of change observed in the fourth stage of discharge.

## 4.4 Comparison with existing literature and method limitations

Compared to conventional needle-plate electrode models used in studies of DC electrical treeing in XLPE insulation, the method proposed in this paper uses a cable structure with real protrusion defects to better simulate field operation

scenarios. This experimental configuration allows for more realistic electric field distributions and partial discharge paths, as seen in [14,17–19]. While needle-plate models are convenient for reproducible laboratory testing, they often fail to reflect the influence of cable geometry, semiconductive shielding, and defect material characteristics on discharge evolution.

Additionally, the combination of experimental discharge data and hydrodynamic simulations based on electron avalanche theory provides insight into the microscopic discharge process and its correlation with macroscopic discharge characteristics, which is not commonly addressed in literature [25–27]. This multi-physics modeling approach offers a more comprehensive understanding of insulation degradation, beyond the phenomenological analysis found in many previous studies.

However, this study also has limitations. The temperature conditions were fixed at 50 °C, which does not fully reflect the dynamic thermal variations experienced by cables in service. Moreover, the discharge behavior under long-term cyclic voltage application or mechanical stress was not explored. From a modeling perspective, the assumptions in the plasma simulation may limit its applicability under certain field conditions. Future work should address these factors and validate the results under broader operating conditions.

## 5. Conclusions

This study investigates the evolution of partial discharge characteristics in cross-linked polyethylene cables with protrusion defects under long-term applied DC voltage. The results highlight the significant role of partial discharge in deteriorating the insulation of XLPE cables, thereby reducing their service life. By analyzing the evolution of discharge quantity and amplitude, the development of defects within the cable insulation can be inferred. The findings suggest that the insulation performance remains stable when discharge amplitude shows little variation. However, a rapid increase in cumulative discharge and discharge amplitude indicates the progression of the protrusion defect, particularly in the fourth stage of discharge, which marks a critical point of failure risk.

A more rigorous, quantitative assessment of partial discharge characteristics reveals that when both cumulative discharge and discharge amplitude rise significantly, the cable is likely to be approaching failure. Future studies could focus on establishing numerical thresholds for these changes to provide clearer diagnostic markers for cable health. Additionally, further simulation and experimental validation are necessary to refine these predictive models and ensure their applicability under various operational conditions.

The insights gained from this study also open the door for extending the analysis to other types of defects, such as air gap and fouling defects, which may have different discharge behaviors. The interaction of these defects with the electric field in the insulation could alter the discharge dynamics, and understanding this relationship is crucial for improving diagnostic accuracy and extending cable lifespan.

Furthermore, the study underscores the importance of early detection techniques for partial discharge and suggests the development of more advanced monitoring systems that could detect subtle changes in discharge characteristics before catastrophic failure occurs. The integration of these diagnostic tools could play a key role in enhancing the maintenance and reliability of high-voltage direct current (HVDC) cable systems, especially in critical infrastructure applications.

This paper demonstrates that partial discharge analysis is a valuable tool for assessing the condition of XLPE cables with protrusion defects. Further work in this field should aim to refine defect models, enhance diagnostic methodologies, and explore mitigation strategies to prevent cable damage, thereby improving the safety and longevity of HVDC cable networks.

## Author contributions

**Conceptualization:** Shuting Yang.

**Formal analysis:** Baiyu Li.

**Methodology:** Jiamin Xu.

**Software:** Baiyu Li.

**Supervision:** Yunjie Zhou.

**Visualization:** Haosheng Lv.

**Writing – original draft:** Baiyu Li.

**Writing – review & editing:** Jialiang Yuan.

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
