## [Decision Letter · Decision Letter 0]

PONE-D-24-49770Analysis of the deterioration process of DC XLPE cable with protrusion defect based on the development of partial dischargePLOS ONE

Dear Dr. Lee,

Thank you for submitting your manuscript to PLOS ONE. After careful consideration, we feel that it has merit but does not fully meet PLOS ONE’s publication criteria as it currently stands. Therefore, we invite you to submit a revised version of the manuscript that addresses the points raised during the review process.

We look forward to receiving your revised manuscript.

Kind regards,

Akhtar Rasool, Ph.D.

Academic Editor

PLOS ONE

**Journal Requirements:**

2. Thank you for stating the following financial disclosure: [This project is supported by the Science and Technology Project of Shanghai Electric Power Company of State Grid, grant number 52090W230008]. Please state what role the funders took in the study. If the funders had no role, please state: "The funders had no role in study design, data collection and analysis, decision to publish, or preparation of the manuscript." If this statement is not correct you must amend it as needed. Please include this amended Role of Funder statement in your cover letter; we will change the online submission form on your behalf.

3. We note that your Data Availability Statement is currently as follows: [All relevant data are within the manuscript and its Supporting Information files.] Please confirm at this time whether or not your submission contains all raw data required to replicate the results of your study. Authors must share the “minimal data set” for their submission. PLOS defines the minimal data set to consist of the data required to replicate all study findings reported in the article, as well as related metadata and methods (https://journals.plos.org/plosone/s/data-availability#loc-minimal-data-set-definition). For example, authors should submit the following data: - The values behind the means, standard deviations and other measures reported; - The values used to build graphs; - The points extracted from images for analysis. Authors do not need to submit their entire data set if only a portion of the data was used in the reported study. If your submission does not contain these data, please either upload them as Supporting Information files or deposit them to a stable, public repository and provide us with the relevant URLs, DOIs, or accession numbers. For a list of recommended repositories, please see https://journals.plos.org/plosone/s/recommended-repositories. If there are ethical or legal restrictions on sharing a de-identified data set, please explain them in detail (e.g., data contain potentially sensitive information, data are owned by a third-party organization, etc.) and who has imposed them (e.g., an ethics committee). Please also provide contact information for a data access committee, ethics committee, or other institutional body to which data requests may be sent. If data are owned by a third party, please indicate how others may request data access.

In addition to the following comments, English needs to be improved particularly punctuation errors need to be fixed.

**Additional Editor Comments:**

Should protrusion be defined concisely with minimum words, before it is taken into consideration?

"partial discharge experiments are carried out at 50 ℃ and 80 kV for internal semi-conductive protrusion defects in insulation in this paper"

This statement is either need to be rephrased to fit to the abstract or be properly supported by the reference or standard (IEEE/IEC), relevant to it. The current form is not proper at all.

Further, the mentioning of the fourth stage in the following statement is also unsupported by any background, this should at least be supported with a short sentence leading to the justification of the mentioning of the fourth stage in the abstract.

"The cumulative discharge has an obvious rapid rising trend, especially in the fourth stage of discharge."

The literature is very limited, given to the fact that a vast material has been published by the filed engineers and the consultants have been published on this subject. So, I see the coverage of the literature needs to be increased. Also, the references, and discussion on relevant IEEE/IEC references be added in the literature, which should lead to the pros and cons of the existing methods/standard listed/published.

I also feel the common kinds of the cable defects be listed, be compared and then protrusion should be lifted based on a small discussion/reasons for choosing in your studies.

Literature review may be presented in tabular form on intrusion defects, improvements, methods, pros and cons leading to the methodology adopted in this current work.

Methodology is good, but I strongly recommend to add a flow chart showing steps of the methodology adopted.

Is there any cross-validation of the results?

Reviewers' comments:

Reviewer's Responses to Questions

**Comments to the Author**

1. Is the manuscript technically sound, and do the data support the conclusions?

Reviewer #1: Yes

Reviewer #2: Yes

2. Has the statistical analysis been performed appropriately and rigorously? 

Reviewer #1: Yes

Reviewer #2: N/A

3. Have the authors made all data underlying the findings in their manuscript fully available?

Reviewer #1: No

Reviewer #2: Yes

4. Is the manuscript presented in an intelligible fashion and written in standard English?

Reviewer #1: No

Reviewer #2: Yes

5. Review Comments to the Author

Reviewer #1: 1. The paper lacks in major contribution and motivation of research.

2. The background and significance of this study should be highlighted in the abstract.

3. Check the English presentation of this paper to remove the typo mistakes. Some grammatical issues need to be addressed in the whole text. Please reform the long paragraphs. Please polish the writing and English of the manuscript carefully. The writing of the paper needs a lot of improvement in terms of grammar, spelling, and presentation. The paper needs careful English polishing since there are many typos and poorly written sentences. I found several errors.

4. In the "Introduction" section, a more detailed analysis of the existing literature on the subject is needed, and an in-depth analysis of the possible application fields.

5. The mathematics used throughout the article is still not very strict. Please try to update and illustrate some elements in the mathematical model that are not defined very strictly.

6. Please categorize your and previous research in the Table in the section

Literature Review to show the better research gap.

7. The overall structure of the article should be improved.

8. The result part is week, results and discussion should be better explained.

9. References must be updated.

10. Check all of your Figures and Tables have a good explanation of your text.

11. Many paragraphs without citations.

12. What are the contributions and novelty of work mentioned?

13. The authors' conclusions need to be improved, a comparison of the results obtained with those already existing in the literature would be appropriate. I suggest also describing what can still be improved in this work, which can still be improved based on the results obtained, according to the authors' view. It is suggested to offer some limitations existed in this study and an outlook for future study in the last section.

Reviewer #2: This manuscript proposes an analysis of the deterioration process of DC XLPE cable with protrusion defect based on the development of partial discharge. The idea is feasible, and there is scientific soundness in this paper. This is one of the good papers I have ever reviewed. However, there are some comments that need to be considered. The reviewer has the following concerns after careful reading:

1. The index terms are not alphabetically ordered.

2. The "Introduction" section should be made much more impressive and enriched with references by adding and citing the trends in the area of the presented method.

3. The paper's classification of context is not addressed at the end of the introduction.

4. In line 91, you mentioned “the total length of the cable is 45 cm," while it is in Fig. 2.a 25 cm.

5. In line 103, you mentioned that “C voltage is rectified by a high-voltage silicon stack, filtered through a 0.1-μF stabilizing capacitor." There is a way of utilizing the line leakage resistance for better filtering. Moreover, the frequency range you mentioned in line 111 is a source of harmonics. Actually, the switching harmonics are a big issue, which you did not consider in the paper. So, this section should be made much more impressive and enriched with references by adding and citing the trends in the area of harmonics mitigation by passive power filters. E.g., High-frequency harmonics suppression in high-speed railway through magnetic integrated LLCL filter; Magnetic integrated double-trap filter utilizing the mutual inductance for reducing current harmonics in high-speed railway traction inverters; Magnetic integrated LLCL filter with resonant ‎frequency above Nyquist frequency; and Enhancing power quality of high-speed railway traction converters by fully integrated T-LCL filter. ‎ Some of these studies, which are published in the same journal, PLOS ONE, are recent and discuss the latest techniques.

6. There are a few writing issues. Also, the correct punctuation rules are not observed in the text.

7. Many figures and equations were not mentioned in the main text or mentioned after they were presented. You need to mention them first.

8. You need to compare the ability of the proposed method with the literature to show what the benefit of the proposed technique is. Moreover, you have to mention the limitations of the proposed technique.

9. Could you insert any reference for equations? It is more suitable to determine the references or explain the design procedure that was adopted to get these equations.

10. The conclusion should be point-wise.

11. In the technical writing, it is not true to write “we, I, our, etc.”.

12. Some of the references are old. A state-of-the-art (SotA) review is preferred to show the real advantages of the proposed method against the recent literature. All the references have no DOI. Many references are conference papers.

In the reviewers' opinion, the presented paper is sufficiently prepared as a professional research paper but needs some modifications. I tried to help you collect these comments as highlighted in the attached file. Please consider my specific comments above. ‎ It could be accepted for publication if my specific comments above were considered.

6. PLOS authors have the option to publish the peer review history of their article (what does this mean? ). If published, this will include your full peer review and any attached files.

**Do you want your identity to be public for this peer review?** For information about this choice, including consent withdrawal, please see our Privacy Policy .

Reviewer #1: No

Reviewer #2: No

---

## [Author Response · Author response to Decision Letter 1]

9 Jan 2025

Dear Editors:

We gratefully appreciate both editors and reviewers for their careful reading, helpful comments, and constructive suggestions, which have significantly improved the quality of our manuscript entitled “Analysis of the deterioration process of DC XLPE cable with protrusion defect based on the development of partial discharge” (Manuscript Number: PONE-D-24-49770).

We have carefully considered all comments from the reviewers and revised our manuscript accordingly. In the following section, we summarize our responses to each comment from the reviewers. We believe that our responses have well addressed all concerns from the reviewers. Here, we would like to re-submit the revised manuscript to PLOS ONE. Hope these revisions could present a better view to the readers and make the paper acceptable for publication in the journal.

Looking forward to hearing from you soon. Have a nice day.

Best regards,

The Authors

Shanghai Jiao Tong University,

Shanghai, 200240, China

+86-18915198870

z1mubaiyu@sjtu.edu.cn

---

## [Decision Letter · Decision Letter 1]

PONE-D-24-49770R1Analysis of the deterioration process of DC XLPE cable with protrusion defect based on the development of partial dischargePLOS ONE

Dear Dr. Lee,

Thank you for submitting your manuscript to PLOS ONE. After careful consideration, we feel that it has merit but does not fully meet PLOS ONE’s publication criteria as it currently stands. Therefore, we invite you to submit a revised version of the manuscript that addresses the points raised during the review process.

We look forward to receiving your revised manuscript.

Kind regards,

Akhtar Rasool, Ph.D.

Academic Editor

PLOS ONE

Journal Requirements:

Additional Editor Comments:

Thanks for addressing the comments of the reviewer in a satisfactory manner. However, one reviewer has asked for some changes yet again. So, it is advised to please address them carefully to avoid any lapse of time in another revision. Thanks

Reviewers' comments:

Reviewer's Responses to Questions

**Comments to the Author**

1. If the authors have adequately addressed your comments raised in a previous round of review and you feel that this manuscript is now acceptable for publication, you may indicate that here to bypass the “Comments to the Author” section, enter your conflict of interest statement in the “Confidential to Editor” section, and submit your "Accept" recommendation.

Reviewer #1: All comments have been addressed

Reviewer #2: All comments have been addressed

2. Is the manuscript technically sound, and do the data support the conclusions?

Reviewer #1: Yes

Reviewer #2: Yes

3. Has the statistical analysis been performed appropriately and rigorously? 

Reviewer #1: Yes

Reviewer #2: N/A

4. Have the authors made all data underlying the findings in their manuscript fully available?

Reviewer #1: Yes

Reviewer #2: No

5. Is the manuscript presented in an intelligible fashion and written in standard English?

Reviewer #1: Yes

Reviewer #2: Yes

6. Review Comments to the Author

Reviewer #1: The authors responded to all my comments. I have no further comments. The paper can be accepted for publication.

Reviewer #2: First of all, I appreciate the authors for the perfect responses and improvements. I am delighted to inform you that this manuscript is very strong now, and I believe that this paper will gain a high number of citations in the future. The authors made a major correction from the previous round. However, this paper requires small corrections because some corrections from the previous round were not done. Kindly find the following notes for further improvements:

1. In the previous round, comment 3 was “The paper's classification of context is not addressed at the end of the introduction.” You have replied to this comment clearly and showed the revision that you have done in the revised manuscript. What I meant is that it is known that at the end of the introduction to present the outline of the paper, like this example: “The rest of this paper is organized as follows. Section 2 would first show the... The... are presented in Section 3. Section 4 provides... Finally, this paper is concluded in Section 5.” So, you may consider this way.

2. In the previous round, comment 8 was, “You need to compare the ability of the proposed method with the literature to show what the benefit of the proposed technique is. Moreover, you have to mention the limitations of the proposed technique.” You have replied to this comment clearly. However, you did not show the revision that you have done in the revised manuscript.

3. In the previous round, comment 11 was, “In the technical writing, it is not true to write “we, I, our, etc.”.” You have replied to this comment clearly and showed the revision that you have done in the revised manuscript. However, there are still some places that were not corrected, or some places in the new content were not written considering this comment.

Thank you for the revisions. The revised manuscript can be accepted for publication if these three minor notices were considered in the final version. The paper is not required to be sent again to the reviewer. You just consider these notes before publishing the paper.

7. PLOS authors have the option to publish the peer review history of their article (what does this mean? ). If published, this will include your full peer review and any attached files.

**Do you want your identity to be public for this peer review?** For information about this choice, including consent withdrawal, please see our Privacy Policy .

Reviewer #1: No

Reviewer #2: No

---

## [Author Response · Author response to Decision Letter 2]

25 Mar 2025

Dear reviewer, hello. In your feedback, it was mentioned that not all the original data was uploaded in this article. According to your request, the original data has been uploaded separately. All the image programs have been uploaded separately, classified as' other ', and these data have been packaged into compressed files named' data 'and uploaded to the system. All the data that can be provided in this article is here. Some of the data was uploaded in the form of screenshots due to simulation reasons. We hope you can understand. Thank you! Hope to receive your reply soon!

---

## [Decision Letter · Decision Letter 2]

PONE-D-24-49770R2Analysis of the deterioration process of DC XLPE cable with protrusion defect based on the development of partial dischargePLOS ONE

Dear Dr. Lee,

Thank you for submitting your manuscript to PLOS ONE. After careful consideration, we feel that it has merit but does not fully meet PLOS ONE’s publication criteria as it currently stands. Therefore, we invite you to submit a revised version of the manuscript that addresses the points raised during the review process.

The response to the prior comment on comparing its results with existing literature and acknowledging limitations (comment 8 from the previous round) is incomplete. 

There are some other minor corrections which need to be done, to help the paper move to the next stage. So, it is kindly requested to address the reviewer suggestions and submit the corrected version at your earliest convenience. 

We look forward to receiving your revised manuscript.

Kind regards,

Akhtar Rasool, Ph.D.

Academic Editor

PLOS ONE

Journal Requirements:

Additional Editor Comments:

The response to the prior comment on comparing its results with existing literature and acknowledging limitations (comment 8 from the previous round) is incomplete.

There are some other minor corrections which need to be done, to help the paper move to the next stage. So, it is kindly requested to address the reviewer suggestions and submit the corrected version at your earliest convenience.

Reviewers' comments:

Reviewer's Responses to Questions

**Comments to the Author**

1. If the authors have adequately addressed your comments raised in a previous round of review and you feel that this manuscript is now acceptable for publication, you may indicate that here to bypass the “Comments to the Author” section, enter your conflict of interest statement in the “Confidential to Editor” section, and submit your "Accept" recommendation.

Reviewer #2: (No Response)

2. Is the manuscript technically sound, and do the data support the conclusions?

Reviewer #2: Yes

3. Has the statistical analysis been performed appropriately and rigorously? 

Reviewer #2: N/A

4. Have the authors made all data underlying the findings in their manuscript fully available?

Reviewer #2: Yes

5. Is the manuscript presented in an intelligible fashion and written in standard English?

Reviewer #2: Yes

6. Review Comments to the Author

Reviewer #2: The authors made a major correction from the previous round. However, this paper requires small corrections because some corrections from the previous round were not done. Kindly find the following notes for further improvements:

1. In the previous round, comment 2 was, (In the previous round, comment 8 was, “You need to compare the ability of the proposed method with the literature to show what the benefit of the proposed technique is. Moreover, you have to mention the limitations of the proposed technique.” You have replied to this comment clearly. However, you did not show the revision that you have done in the revised manuscript.). You added this paragraph in Section 2.4, Page 8 “To provide a clearer understanding of the logical flow of this article, a flowchart is presented below. This flowchart summarizes the overall logical structure and progression of the content: it begins with the introduction section, which outlines the research background and issues, as well as clarifies the research objectives; next, the methods section elaborates on the research design and experimental approaches; then, the results section presents the analyzed data and key findings; following that, the discussion section interprets the significance of the research results and compares them with existing literature; finally, the conclusion section summarizes the research contributions and offers prospects for future work.”, which is a repeat of the paper's classification of context at the end of the introduction, not the required correction. You need to reconsider this comment.

2. In the previous round, comment 3 was, (“In the technical writing, it is not true to write “we, I, our, etc.”.” You have replied to this comment clearly and showed the revision that you have done in the revised manuscript. However, there are still some places that were not corrected, or some places in the new content were not written considering this comment.) You made some modifications but some of them are not correct. Specifically, in Introduction Section, Page 2 “To address this, by investigating the behavior of protrusion defects in DC cables under controlled voltage and temperature conditions, monitoring the variation in PD characteristics.” And in Conclusion Section, Page 17, “By analyzing the evolution of discharge quantity and amplitude, it can be inferred that the development of defects within the cable insulation.” These two sentences are not grammatically true.

The revised manuscript can be accepted for publication if these two minor notices were considered in the final version. The paper is not required to be sent again to the reviewer. You just consider these notes before publishing the paper.

7. PLOS authors have the option to publish the peer review history of their article (what does this mean? ). If published, this will include your full peer review and any attached files.

**Do you want your identity to be public for this peer review?** For information about this choice, including consent withdrawal, please see our Privacy Policy .

Reviewer #2: No

---

## [Author Response · Author response to Decision Letter 3]

14 May 2025

Revision Report for Manuscript PONE-D-24-49770R2

Dear Editors:

We gratefully appreciate both editors and reviewers for their careful reading, helpful comments, and constructive suggestions, which have significantly improved the quality of our manuscript entitled “Analysis of the deterioration process of DC XLPE cable with protrusion defect based on the development of partial discharge” (Manuscript Number: PONE-D-24-49770R2).

We have carefully considered all comments from the reviewers and revised our manuscript accordingly. In the following section, we summarize our responses to each comment from the reviewers. We believe that our responses have well addressed all concerns from the reviewers. Here, we would like to re-submit the revised manuscript to PLOS ONE. Hope these revisions could present a better view to the readers and make the paper acceptable for publication in the journal.

Looking forward to hearing from you soon. Have a nice day.

Best regards,

The Authors

Shanghai Jiao Tong University,

Shanghai, 200240, China

+86-18915198870

z1mubaiyu@sjtu.edu.cn

Response to Reviewer #2:

Specific comments 1:

1. In the previous round, comment 2 was, (In the previous round, comment 8 was, “You need to compare the ability of the proposed method with the literature to show what the benefit of the proposed technique is. Moreover, you have to mention the limitations of the proposed technique.” You have replied to this comment clearly. However, you did not show the revision that you have done in the revised manuscript.). You added this paragraph in Section 2.4, Page 8 “To provide a clearer understanding of the logical flow of this article, a flowchart is presented below. This flowchart summarizes the overall logical structure and progression of the content: it begins with the introduction section, which outlines the research background and issues, as well as clarifies the research objectives; next, the methods section elaborates on the research design and experimental approaches; then, the results section presents the analyzed data and key findings; following that, the discussion section interprets the significance of the research results and compares them with existing literature; finally, the conclusion section summarizes the research contributions and offers prospects for future work.”, which is a repeat of the paper's classification of context at the end of the introduction, not the required correction. You need to reconsider this comment.

Answer:

We sincerely thank the reviewer for this important reminder. In the previous revision, we inadvertently misunderstood the intent of the comment and added only a general flowchart in Section 2.4, which repeated the structural overview from the Introduction and did not constitute a valid response.

In this revised version, we have now explicitly added a new subsection titled “4.4 Comparison with existing literature and method limitations” (page 17), in which we:

Compare our method (cable-based protrusion defect model with combined experiment and simulation) with traditional approaches (needle-plate models),

Highlight its advantages in representing realistic field scenarios and capturing both macroscopic and microscopic discharge behaviors,

Acknowledge limitations regarding thermal condition control, lack of cyclic voltage or mechanical stress testing, and simplifications in the simulation model.

We hope this revision fully addresses the reviewer’s concern.

Revision:

Manuscript Section

Page 17

4.4 Comparison with existing literature and method limitations

Compared to conventional needle-plate electrode models used in studies of DC electrical treeing in XLPE insulation, the method proposed in this paper uses a cable structure with real protrusion defects to better simulate field operation scenarios. This experimental configuration allows for more realistic electric field distributions and partial discharge paths, as seen in [14, 17–19]. While needle-plate models are convenient for reproducible laboratory testing, they often fail to reflect the influence of cable geometry, semiconductive shielding, and defect material characteristics on discharge evolution.

Additionally, the combination of experimental discharge data and hydrodynamic simulations based on electron avalanche theory provides insight into the microscopic discharge process and its correlation with macroscopic discharge characteristics, which is not commonly addressed in literature [25–27]. This multi-physics modeling approach offers a more comprehensive understanding of insulation degradation, beyond the phenomenological analysis found in many previous studies.

However, this study also has limitations. The temperature conditions were fixed at 50 °C, which does not fully reflect the dynamic thermal variations experienced by cables in service. Moreover, the discharge behavior under long-term cyclic voltage application or mechanical stress was not explored. From a modeling perspective, the assumptions in the plasma simulation (e.g., neglecting space charge accumulation or surface charge relaxation dynamics) may limit its applicability under certain field conditions. Future work should address these factors and validate the results under broader operating conditions.

Specific comments 2:

2. In the previous round, comment 3 was, (“In the technical writing, it is not true to write “we, I, our, etc.”.” You have replied to this comment clearly and showed the revision that you have done in the revised manuscript. However, there are still some places that were not corrected, or some places in the new content were not written considering this comment.) You made some modifications but some of them are not correct. Specifically, in Introduction Section, Page 2 “To address this, by investigating the behavior of protrusion defects in DC cables under controlled voltage and temperature conditions, monitoring the variation in PD characteristics.” And in Conclusion Section, Page 17, “By analyzing the evolution of discharge quantity and amplitude, it can be inferred that the development of defects within the cable insulation.” These two sentences are not grammatically true.

The revised manuscript can be accepted for publication if these two minor notices were considered in the final version. The paper is not required to be sent again to the reviewer. You just consider these notes before publishing the paper.

Answer:

We thank the reviewer for carefully identifying these remaining issues. In the current revision, we have corrected the grammatical errors and ensured alignment with academic writing standards. The specific changes are as follows:

Revision:

Section 1

Page 2

Most studies on PD in XLPE cables under DC voltage focus on pin-plate electrode configurations, which are insufficient to model real-world cable defects [17,18]. Furthermore, the discharge channels are difficult to observe directly in situ, though PD characteristics remain strongly correlated with their development [19]. To address this issue, the behavior of protrusion defects in DC cables was investigated under controlled voltage and temperature conditions, and the variation in partial discharge (PD) characteristics was monitored. A finite element simulation model is established to simulate the electric field distribution and the development of discharge channels, based on electron avalanche theory. This model helps to explore the relationship between electric field distortion at the defect site, discharge magnitude, and channel size. The study aims to provide a new method for monitoring the stages of defect progression, offering insights into the early detection of faults in HVDC cables.

Section 5

Page 18

This study investigates the evolution of partial discharge (PD) characteristics in cross-linked polyethylene (XLPE) cables with protrusion defects under long-term applied DC voltage. The results highlight the significant role of partial discharge in deteriorating the insulation of XLPE cables, thereby reducing their service life. By analyzing the evolution of discharge quantity and amplitude, the development of defects within the cable insulation can be inferred. The findings suggest that the insulation performance remains stable when discharge amplitude shows little variation. However, a rapid increase in cumulative discharge and discharge amplitude indicates the progression of the protrusion defect, particularly in the fourth stage of discharge, which marks a critical point of failure risk.

---

## [Decision Letter · Decision Letter 3]

Analysis of the deterioration process of DC XLPE cable with protrusion defect based on the development of partial discharge

PONE-D-24-49770R3

Dear Dr. Lee,

We’re pleased to inform you that your manuscript has been judged scientifically suitable for publication and will be formally accepted for publication once it meets all outstanding technical requirements.

Kind regards,

Akhtar Rasool, Ph.D.

Academic Editor

PLOS ONE

Additional Editor Comments (optional):

Congratulations

Reviewers' comments:

Reviewer's Responses to Questions

**Comments to the Author**

1. If the authors have adequately addressed your comments raised in a previous round of review and you feel that this manuscript is now acceptable for publication, you may indicate that here to bypass the “Comments to the Author” section, enter your conflict of interest statement in the “Confidential to Editor” section, and submit your "Accept" recommendation.

Reviewer #2: All comments have been addressed

2. Is the manuscript technically sound, and do the data support the conclusions?

Reviewer #2: Yes

3. Has the statistical analysis been performed appropriately and rigorously? 

Reviewer #2: N/A

4. Have the authors made all data underlying the findings in their manuscript fully available?

Reviewer #2: Yes

5. Is the manuscript presented in an intelligible fashion and written in standard English?

Reviewer #2: Yes

6. Review Comments to the Author

Reviewer #2: Thank you for the revisions. The revised manuscript can be accepted for publication. Congratulations.

7. PLOS authors have the option to publish the peer review history of their article (what does this mean? ). If published, this will include your full peer review and any attached files.

**Do you want your identity to be public for this peer review?** For information about this choice, including consent withdrawal, please see our Privacy Policy .

Reviewer #2: No

---

## [Editor Report · Acceptance letter]

PONE-D-24-49770R3

PLOS ONE

Dear Dr. Li,

I'm pleased to inform you that your manuscript has been deemed suitable for publication in PLOS ONE. Congratulations! Your manuscript is now being handed over to our production team.

Kind regards,

on behalf of

Dr. Akhtar Rasool

Academic Editor

PLOS ONE